# Recent Advances in Temporomandibular Joint Surgery

**DOI:** 10.3390/medicina59081409

**Published:** 2023-08-02

**Authors:** Catherine Wroclawski, Jai Kumar Mediratta, W. Jonathan Fillmore

**Affiliations:** 1Resident, Oral and Maxillofacial Surgery, Mayo Clinic, Rochester, MN 55905, USA; 2Consultant, Oral and Maxillofacial Surgery, Mayo Clinic, Rochester, MN 55905, USA

**Keywords:** temporomandibular joint, TMJ, surgery, arthroscopy, arthroplasty, dislocation, eTMJR

## Abstract

Temporomandibular disorders (TMDs) affect a high percentage of children and adults worldwide. Surgery may be indicated in severe or recalcitrant cases. Several recent advancements in TMD and temporomandibular joint (TMJ) surgery have elevated understanding and the ability to treat affected patients. We discuss recent advances in TMD epidemiology, juvenile idiopathic arthritis (JIA) of the TMJ, and surgical techniques and technologies. Technical advancements have been identified in TMJ arthroscopy, the treatment of TMJ subluxation and dislocation, and extended prosthetic total TMJ reconstruction (eTMJR). Overall, this review provides valuable insights into significant recent advancements in TMJ disorders and their surgical management.

## 1. Introduction

Temporomandibular disorders of all varieties affect people in every culture on every continent, affecting up to 11% of children and even as high as 31% of adults [1]. While most patients experience relief or at least a significant improvement in symptoms with noninvasive treatments, surgical care may be warranted in refractory cases or frank arthritic disease, among other scenarios.

A surgeon’s management of temporomandibular disorders has progressed over the last several years. This is due, in large part, to advances in understanding pathophysiology, epidemiology, and both surgical techniques and technologies. We may often look to new technology in innovation for improved outcomes for patients. While this is the case for TMJ surgery, advances in the foundational understanding of the disease state have also brought significant benefits to patients and surgeons alike, leading to better outcomes through better patient selection for surgical intervention, among other modes of care. The purpose of this article is to review recent major advances that will assist the practicing TMJ surgeon in providing high-quality, current, and informed care.

A basic understanding of TMD-related epidemiology has helped surgeons appreciate and apply more holistic and collaborative care models as well as recognize patient cohorts who may be more likely to experience favorable outcomes through surgical intervention. Similarly, the last several years have seen a renewed understanding of TMJ involvement in patients with juvenile idiopathic arthritis (JIA). Through improved understanding, consensus diagnostics, imaging, and treatment protocols aim to improve patient outcomes. On the technical side, arthroscopic TMJ surgery has also seen a proliferation of advocates and techniques, particularly with regard to arthroscopic disc repositioning. Additionally, recent investigations have helped identify additional treatments for recurrent TMJ dislocation. Finally, while prosthetic TMJ reconstruction (TMJR) has been firmly established as a safe and effective technique for the treatment of many severe conditions [2], new frontiers of the application of this technology have been opened through the use of extended TMJR prostheses.

While not exhaustive, these areas of recent progress represent important advances for surgeons treating the TMJ, about which they should certainly be aware to provide current and optimal care for their patients (Figure 1).

## 2. TMD Epidemiology: The OPPERA Study

The Orofacial Pain: Prospective Evaluation and Risk Assessment (OPPERA) study is a landmark longitudinal study of temporomandibular disorders (TMDs), with its first publication beginning in 2011 [3]. It aimed to improve the understanding of orofacial pain and TMD by identifying risk factors, clarifying pain mechanisms, assessing treatment outcomes, and examining psychosocial contributions to the disease, among other metrics. Unlike previous studies that primarily used cross-sectional designs and convenience samples, OPPERA overcame this limitation by employing a prospective cohort study with a multi-site approach. It has provided valuable insights into TMD prevalence, demographic trends, the role of jaw headaches and injury as risk factors, and the differentiation between primary headaches and TMD-related secondary headaches, among other findings. Furthermore, the study has facilitated the identification of phenotypical clusters, allowing for a more targeted and comprehensive approach to TMD management in clinical practice. While well known among those involved in headache and facial pain treatment, the authors believe these studies are vastly underrecognized and underutilized in the surgical community.

The OPPERA-I study, conducted between May 2006 and May 2013, included a total of 4346 subjects with an age range of 18–44 living in Baltimore, MD; Buffalo, NY; Chapel Hill, NC; and Gainsville, FL, who were studied between 2006 and 2013. There were three study designs utilized: a prospective cohort study, a case-control study, and a nested case-control study. Between December 2014 and May 2016, the OPPERA-2 study was conducted, connecting with subjects who had not previously withdrawn from OPPERA-1. For individuals who provided consent and participated in the research clinics during OPPERA-2, data was gathered through clinical examinations, quantitative sensory testing, blood samples, and self-reported questionnaires.

### 2.1. Demographics

One of OPPERA’s major strengths is its study design. For example, in 1993, Bush et al. identified that TMD has a higher prevalence in females than in males [4], with LeResche further characterizing TMD as affecting young and middle-aged adults in 1997 [5]. However, the initial findings of the OPPERA studies revealed that TMD is prevalent in the US, with 4% of the population developing TMD on an annual basis [6]. Slade et al. also published the initial findings of the OPPERA study in 2011 revealing the key demographic data [3]. TMD is highest in patients aged 35–44 (7%) and lowest in patients aged 18–24 (3%). Females are four times more likely to suffer from TMD than men, while African Americans and Hispanics had one-fifth odds of developing TMD compared to non-Hispanic Whites. However, Kim et al. performed a cross-sectional study utilizing the OPPERA database, showing that for both Asians and African Americans compared to non-Hispanic Whites, pain catastrophizing played a significant role in mediating the association between race and pain-related measures [7]. Further, compared to healthy individuals, patients with TMD have higher levels of pain-related disability, and it is well established that catastrophizing is a strong predictor of TMD, as are psychosocial stress, affective distress, and somatic symptoms overall [8]. It is crucial to consider these factors appropriately when treatment planning potential surgical candidates.

### 2.2. Role of Injury

The OPPERA study has played a pivotal role in our understanding of the development and exacerbation of TMD. Sharma et al. in 2019 sought to elucidate jaw injury as a risk factor for TMD [9]. The initial findings of the prospective cohort study showed that individuals who reported a jaw injury had a fourfold higher risk of developing TMD compared to those without an injury, either extrinsic or intrinsic. Intrinsic was defined as injury attributed to yawning or prolonged mouth opening, whereas extrinsic included tooth extraction or dental treatments; oral intubation; sports injury (including falls, bumps, and blows); motor vehicle accidents; accidents resulting in whiplash; and injuries to the shoulder, neck, and head region. Sharma et al. would expand upon their initial study to show that jaw injury significantly increased the odds of developing TMD by a factor of five after adjusting for confounding variables [10]. In both studies performed, intrinsic injuries were more prevalent than extrinsic. Despite most injuries being intrinsic, the risk of TMD increased to a similar extent irrespective of whether the injury was intrinsic or extrinsic.

### 2.3. Headache and Chronic Pain Conditions

The OPPERA studies shed light on the association between headaches and their role as a risk factor for TMD. Headaches that arise as a secondary symptom of TMD are distinct from primary headaches that can subsequently contribute to the development of TMD. Headaches attributed to TMD (HATMD) are well established and part of Schiffman’s diagnostic criteria for TMD [11] and was recently included in the finalized version of the International Classification of Headache Disorders—third edition (ICHD-3) [12]. In contrast, primary headaches, such as migraines or tension headaches, are known to have an independent etiology but are thought to be comorbid with TMD. Tchivileva et al. utilized the OPPERA data to show that baseline reports of migraine, as well as frequent headaches, served as risk factors for developing TMD [13]. The association between migraines and TMD is unsurprising when considering the shared anatomical pathway of the trigeminal nerve between migraines and TMD [14]. Tchivileva et al. later showed that not only is the baseline presentation of a migraine a risk factor but that HATMD typically presents as migraine dominating. Sonia et al. sought to define characteristics that distinguish a headache that is comorbid with temporomandibular disorder from a HATMD. Utilizing the data from the OPPERA II study, they showed that the more severe the masticatory system pain with a patient, the more likely the headache was secondary to HATMD versus a primary headache [15].

Headache was not the only pain condition studied within the OPPERA data set. Other chronic overlapping pain conditions (COPCs) often coexist with TMD and exhibit shared biopsychosocial characteristics, symptoms, and risk factors [16]. Research has shown that the overlapping of disease symptoms of COPC can be attributed to central sensitization, a mechanism involving increased synaptic efficiency that affects sensory and nociceptive stimuli [17]. Slade et al. utilized the OPPERA II data set to compare the degree of overlap between TMD and other COPCs, revealing a greater overlap between musculoskeletal conditions such as fibromyalgia and chronic back pain compared to headaches and IBS [18].

### 2.4. Phenotypical Clusters

While the OPPERA studies have shed light on the heterogenous etiology of TMD with an anatomical focus, they have also statistically validated several risk factors seen among all chronic pain patients allowing for the development of “phenotypical clusters.” Bair et al. were the first to describe these clusters by identifying statistically distinguishable risk factor profiles that patients shared [16]. The three phenotypical clusters described were the adaptive cluster, pain-sensitive cluster (PS), and global symptoms (GS) cluster based on four validated variables: the pressure pain threshold of the trapezius muscle and its association with the subscales of anxiety, depression, and somatization, as measured by the SCL-90R questionnaire [19].

In this study, 33% belonged to the adaptive cluster, which showed minimal hypersensitivity and psychological distress. Individuals in this particular group experience milder pain, minimal psychological distress, and exhibit the lowest sensitivity to muscle pain. The pain they experience is primarily confined to the temporomandibular joint and the surrounding muscles, and they report few comorbidities or COPCs. Additionally, this group has a higher representation of men compared to women. The pain-sensitive cluster comprised 48% of participants. Individuals in this group exhibit the highest sensitivity to muscle pain, slightly elevated psychological distress, and a greater number of associated chronic pain conditions compared to the adaptive group. This group also has a slightly higher number of women compared to men. The remaining cluster, known as global symptoms, accounted for 18% of the participants in Bair’s study. Individuals in this group who have TMD experience the most intense pain and dysfunction. They exhibit a higher number of tender muscle sites and a significantly greater number of associated COPCs. Moreover, they encounter the highest levels of psychological distress among the three groups. Individuals in this group tend to have a higher incidence of jaw injuries and have experienced more traumatic events in their lives. A larger percentage of patients in this group have a history of smoking [19]. Lastly, there is a notable imbalance in gender representation, with a considerably higher proportion of women than men falling into this group.

OPPERA has improved our understanding of the epidemiology of TMD as well as the risk factors associated with the condition. By clarifying the risk factors, such as psychosocial stress, affective distress, somatic symptoms, trauma, and headaches, surgeons can better identify individuals at higher risk and develop targeted prevention strategies. The utilization of the phenotypical clustering method holds great promise for influencing clinical practice in a meaningful way as well. OPPERA has shown that, for individuals in either the PS or GS clusters, it is crucial to initiate a thorough screening for concurrent pain conditions. While there may be clear anatomic abnormalities in a potential TMJ surgical candidate, understanding and applying the findings of the OPPERA studies will help surgeons to make more clear-eyed clinical decisions, seek multidisciplinary treatment, and more thoroughly educate patients to help them understand the implications of their condition. The authors recommend a full review of the OPPERA studies for a more in-depth understanding of the subject and its application.

## 3. Juvenile Idiopathic Arthritis of the Temporomandibular Joint

Understanding and treatment of juvenile idiopathic arthritis (JIA) and its effects, specifically on the temporomandibular joint, have advanced significantly over the last decade. Once noted to be “the forgotten joint”, [20] due to its omission from diagnosis and management protocols for JIA, the last 10–15 years have hosted a dramatic increase in our understanding about TMJ involvement in sufferers of JIA [21,22,23]. This includes improvements in detection and diagnosis, generalized increased awareness, recognition of morbidity of untreated cases, and established medical and surgical treatments in this patient population.

JIA affects 1 in 1000 children [24] and has seven subtypes, according to the International League of Associations for Rheumatology [25]. It appears that TMJ involvement in these cases is very common (up to 87% [26], but more likely close to 40% [27,28]), albeit frequently undetected due to a lack of symptoms or screening.

### 3.1. Imaging and Diagnosis

Physical examination is still important, as is a thorough history, in order to identify potential TMJ involvement [29]. However, it is clear that the exam alone is insufficient for screening in the JIA population. A straightforward and efficient six-point examination protocol has been developed by the TMJaw Working Group [30] but should be supplemented with imaging. It is recommended that patients undergo regular TMJ-focused exams as part of initial rheumatologic/JIA evaluations as well as at the time of annual follow-up in order to determine the presence or progression of disease [31].

Imaging these patients becomes critical in light of the frequent lack of symptoms in the presence of early or even destructive late disease. Fortunately, the development of recent guidelines and protocols [32,33] and resources [34,35] reduces the number of patients missed or undertreated. For proper screening to evaluate synovitis or early joint disease, Gadolinium contrast-enhanced MRI is ideal [36,37]. In addition, the wider availability of 3.0-T magnets in recent years has also enhanced diagnostic ability.

It is important for the treating surgeon to communicate clearly with the radiology team to ensure proper imaging is acquired, especially in the case of children who may require sedation, or in whom an abbreviated exam is desirable. A “minimal required protocol” includes several sequences (sagittal oblique fat-suppressed T2 or STIR, sagittal oblique T1 TSE, coronal T1 TSE, and contrast-enhanced T1 FS-weighted), while a longer “ideal protocol” adds sagittal oblique fat-suppressed T1, proton density, and gradient echo sequences [32]. The use of a scoring system for joint health based on these protocols helps stage disease or monitor progression/quiescence [34,38,39]. Further improvements in imaging techniques, such as dynamic contrast enhancement MRI [40] and black bone MRI [41], potentiate earlier detection and discrimination of disease to allow for more complete and timely treatment of this patient population.

Other means of diagnosis remain unsettled or controversial in the scientific literature at this point. This includes the use of panorex, sonography, and synovial fluid analysis, among other methods [42]. In addition, MRI is helpful to detect inflammation and osseous changes but is not necessarily specific for JIA as the etiology of these findings [43]. CT or cone beam CT may assist in detecting osseous changes or evaluating craniofacial changes in affected individuals, but will be unhelpful in monitoring soft tissue inflammation [44].

### 3.2. Disease Course and Treatment

With greater publicity of TMJ involvement in JIA patients, the late sequelae of un- or undertreated disease have become more visible [22]. Likewise, treatment aimed at reducing the incidence and burden of late-stage disease has advanced.

Since the TMJ is a growth area for the mandible and influences craniofacial growth and development beyond just the condyle itself, these disruptions can cause significant hypoplasia of the mandible, malpositioning of the maxilla, and retrognathia-associated airway compromise or sleep-disordered breathing [23,45,46]. The degree and type of dentofacial deformity are related to the timing and severity of the disease during growth and development [47].

Overall, the optimization of medical treatments for JIA seems to benefit affected temporomandibular joints. Advances in systemic treatments are encouraging; however, in many instances, the TMJ seems to respond less well to medical treatments than other joints [20,28]. However, a recent study by Bollhalder et al. showed hopeful TMJ-related outcomes with methotrexate (and, in many cases, combined with biologics) over several years of treatment [48]. Encouraging outcomes included both symptom control and mandibular growth. Fortunately, more studies are forthcoming to better understand the effects of systemic therapy on the TMJ, in particular, including clinical trials [49]. It is unclear if this has more to do with the overall incidence of temporomandibular disorders (TMDs) casting a confounding shadow over concurrent JIA or if there is something different about the TMJ that causes the TMJ to be less responsive to systemic treatment.

Intra-articular injection of steroids has been advocated for a long time (Stoll 2015), but there is some controversy regarding repeated use and the potential risk of heterotopic bone formation or problematic growth [50,51]. It may be helpful for symptom control (not disease progression), but repeated injections of steroids after an unsuccessful injection should not be encouraged [52]. Interestingly, arthrocentesis without steroid injection was shown to be as effective for symptom control as arthrocentesis with steroid injection [53]. Alternative injections, such as inflixamab, have been explored for systemic arthropathy [54] and the TMJ in particular with mixed results [55,56].

For late-stage disease, with or without extra-articular dentofacial deformity, major operative intervention can be quite successful [57]. For the treatment of the joint itself, orthognathic surgery, costochondral grafting, distraction osteogenesis [58], and prosthetic joint reconstruction have been advocated. Costochondral grafting (CCG)—a successful treatment [59]—seems to be less favored recently, as alloplastic TMJ reconstruction is often viewed by surgeons as more predictable biologically and functionally [60]. Early reports that have shown encouraging success in prosthetic joint reconstruction are encouraging in terms of an improved range of motion, occlusal and masticatory function, and even improvement in obstructive sleep apnea (OSA) [61]. This is a particularly valuable treatment when inflammation and synovitis persist despite medical interventions, or when the counterclockwise rotation of the jaws is planned and autogenous grafts may not be as robust mechanically. In cases involving dentofacial deformity, an interdisciplinary approach (involving orthodontists, oral and maxillofacial surgeons, and pediatric rheumatologists) may be especially helpful [62].

Orthognathic surgery, with or without arthroplasty, has a place in the treatment of this patient population. Reporting on combined orthognathic and prosthetic joint reconstruction, Trivedi et al. evaluated 40 JIA-affected patients in a case-control model and showed dramatic improvements in function and a reduction in pain [63]. Regarding mandibular orthognathic surgery, Raffaini et al. reviewed 13 cases retrospectively, noting stable findings and quiescent disease 1 year post-operatively [64]. All of these patients had some condylar changes and reduced ramus height (12 bilateral), but disease was quiescent in the TMJ at the time of surgery. In addition, all underwent medical treatment in the form of etanercept for at least one year prior to surgery. Other reports of orthognathic surgery in this population seem to indicate stable outcomes in the quiescent joint [65,66,67]. Similarly, distraction osteogenesis may be considered in the same population [62].

Another concern for JIA-affected individuals, particularly with TMJ involvement, is the risk of developing obstructive sleep apnea (OSA). Recent studies indicate the significant risk of OSA in this population as compared to non-JIA individuals [68,69]. The condylar resorption and rotation of the mandible that causes retrognathia seem to be a significant factor in developing these symptoms in the young adult years (age 18–30) [68]. This is simply another reason for early detection and intervention in this patient population in order to minimize the risk of dentofacial deformity and all its problematic sequelae.

Finally, treatments commonly used for temporomandibular disorders, such as physical therapy or occlusal appliances, may be implemented for symptom relief [70,71]. If a patient responds to an occlusal appliance alone, for example, it is less likely that the underlying source of symptoms is inflammatory arthritis.

## 4. Technical Advances in TMJ Surgery

### 4.1. Advanced TMJ Arthroscopy

TMJ arthroscopy is a treatment modality with a rich history (Figure 2), and there are several areas within arthroscopy that have seen advancement recently. One of the most notable developments in advanced TMJ arthroscopy in the last 10–15 years has been the development of discopexy for the repositioning and fixation of an anteriorly displaced TMJ disc [72,73]. First reported in English by Israel in 1989 [74], important variations soon followed [75,76,77,78,79]. The technique documented in 1992 by McCain et al. [80], involving 11 temporomandibular joints (8 patients), was subsequently modified by Yang et al. in 2012 [81]. The technique described by McCain et al. involves the release of the anterior portion of the disc from its attachment to the synovium. Once the disc is reduced, it is sutured posterolaterally. With this technique, the suture is passed through the posterior margin of the disc using a spinal needle, while the Meniscus Mender II uses a lasso-type suture retriever. McCain’s technique involves a small incision within the preauricular crease adjacent to the suture exit point to facilitate tying the suture within the extracapsular fatty tissue.

Yang and colleagues’ main modifications to this suture discopexy technique is in the suturing technique and instruments used [81]. Most notably, a horizontal mattress pattern is used with two sutures and the sutures are tied such that the knots are beneath the cartilage of the external auditory canal (EAC). This method of suturing not only prevents dimpling of the skin, but also minimizes the risk of entrapment of the frontal branch of the facial nerve and allows for a vector of traction on the disc that is directed along the anterior–posterior long axis of the disc, as opposed to the posterolateral traction of prior techniques. Yang’s technique also involves an exchangeable, custom-designed lasso-type and hook-type gripper.

A follow-up study by Yang et al. that used MRI to evaluate the efficacy of their arthroscopic suture discopexy technique to reposition anteriorly displaced discs in 764 joints found a success rate of 98.56%. However, the post-operative MRI was obtained only between 1 and 7 days post-operatively [86]. More recently, Jerez et al. describe a modification to Yang’s suture discopexy technique that utilizes more commonly available suture equipment consisting of two patented lasso grippers, and two Meniscus Mender II curved and straight spinal needles. However, this technique requires five to six puncture sites compared to three puncture sites with Yang’s technique [84].

Alternative techniques for discopexy have also arisen. Martinez-Gimeno has described a single portal discopexy technique, fixing the disc to the tragal cartilage without an anterior release [87]. A 1-year follow-up seemed favorable for a limited number of subjects, most with anterior disc displacement with reduction. Alternatively, arthroscopic discopexy using resorbable pins in lieu of sutures has also recently received more long-term appraisal, showing efficacy [88]. Using the technique initially published in 2016 [89], and similar to Goizueta-Adame in 2014 [90], resorbable pin use appears to offer at least five years of benefit, in terms of range of motion and pain reduction. Although the sample consisted of 33 subjects and only 23 made it to 5-year follow-up, the findings are encouraging, and larger case numbers will further solidify this technique as viable for a Wilkes III patient. Lastly, a separate study reported arthroscopic use of a titanium anchor for disc repositioning and 6 months of follow-up with patient benefits in a Wilkes II-III population [91].

Multiple studies have assessed the effectiveness of disc repositioning and suturing in arthroscopic vs. open techniques. A study by Abdelrehem et al. evaluated the outcomes of TMJ arthroscopic versus open disc repositioning for the management of anterior disc displacement in 277 joints (177 patients) [92]. This study found that while there was an improvement in pain score, clicking, diet, and MIO, in both the arthroscopic and open groups, the clinical improvements occurred earlier in the arthroscopic group (1 month) versus the open group (6 months.) Additionally, the success rate in the arthroscopic group was slightly higher than the open group at 98.1% versus 97.3%. Condylar remodeling occurred in 70.2% of patients in the arthroscopy group versus 30.1% of patients in the open group. Recent systematic reviews by Askar et al. and Santos et al. have found that arthroscopic and open disc repositioning reduced pain and improved MIO. However, both studies indicated that the number of studies and evidence was limited, with Askar et al. noting that the heterogeneous nature of the study designs and data reporting was such that the studies could not be directly compared, and quantitative analysis could not be performed [93,94].

The authors also noted the lack of large subject numbers as well as medium and, especially, long-term follow-up for either open or arthroscopic disc repositioning studies. Because of these deficiencies, it is difficult to advocate for one technique over another, and more data is needed. While remaining optimistic that this treatment benefits many patients, the current volume and quality of data are consistent with the controversial view this surgery may at times garner. One possible virtue of the arthroscopic technique is the avoidance of open surgery, the accumulation of which often renders a later total joint arthroplasty less successful. However, we also have no helpful data on outcomes upon conversion of arthroscopic disc repositioning to total joint arthroplasty. While less invasive than an open technique, it is unclear if this advanced arthroscopic technique “counts against” the tally of prior TMJ surgeries that would decrease the success of a later prosthetic reconstruction.

The Wilkes classification of TMJ internal derangement has been shown to predict the likelihood of a successful arthroscopic discopexy. McCain et al. found that Wilkes stages II and III had a successful primary outcome (the absence of joint pain at 12 months post-operatively) of 86.7% compared to 25% for Wilkes stages IV and V [85]. This contrasts an older study by Murakami et al. that reported a 92% and 93% success rate for Wilkes stages IV and V, respectively [95]. However, this study utilized arthroscopic lysis and lavage for Wilkes stage IV and advanced arthroscopic procedures (synovectomy, discoplasty, and debridement) for Wilkes stage V, as opposed to discopexy.

Some of the findings regarding Wilkes staging and success in arthroscopic discopexy were further assessed in a very recent study by Sah et al. [96]. This retrospective study assessed whether certain prognostic factors impacted the success of Yang’s arthroscopic suture discopexy technique in the treatment of TMJ closed lock. It was found that age, duration of illness, Wilkes classification, and prior orthodontic treatment all impacted surgical outcomes. Specifically, younger age, Wilkes stage III, shorter duration of illness, and current orthodontic treatment were all associated with positive surgical outcomes. On the contrary, older age, Wilkes stage IV, longer duration of illness, and previous orthodontic treatment were associated with poor surgical outcomes.

When considering the efficacy of TMJ disc repositioning in cases of anterior disc displacement, it is also important to consider the ability of disc repositioning to prevent complications that could arise secondary to a lack of treatment of disc displacement. In the adolescent population, untreated unilateral TMJ anterior disc displacement may result in mandibular asymmetry, while bilateral anterior disc displacement may result in mandibular retrusion. Prior to recently, there were very few studies that evaluated condylar bone remodeling following arthroscopic TMJ surgery. Condylar bone remodeling following the treatment of disc displacement would be beneficial in preventing complications regarding mandibular symmetry and, thus, occlusion in patients with disc displacement. Dong and colleagues recently performed a study to evaluate condylar remodeling following Yang’s arthroscopic surgery in patients with anterior disc displacement both with and without reduction [83]. While the specifics of the arthroscopic surgery patients underwent were not discussed in detail, they found that 70.3% of the 229 patients had new condylar bone formation when evaluated with MRI at 1 year following their arthroscopic surgery. The youngest age group (10–15 years old) had the greatest percentage of patients with new condylar bone formation (94.33%), while the oldest age group (above 30 years old) had the lowest percentage (25%). The percentage of new condylar bone formation was found to decrease as patient age increased [83]. More than half of the patients (53.53%) had bone formation on the posterior slope of the condyle. Meanwhile, the area of the condyle with the smallest amount of bone formation was the anterior slope of the condyle, with only 10.37% of patients having new bone formation in this area. This study is important in showing that the condyle still has the propensity to form new bone, especially in younger patients, after repositioning of the TMJ disc. This may ultimately represent a protective factor in preventing mandibular asymmetry and retrusion in cases of anterior disc displacement.

Outside of arthroscopic discopexy, arthroscopic management of a painful or problematic alloplastic TMJ prosthesis has been recently reported [97]. This is an important advancement in technique, as TMJ prostheses are gaining traction and becoming more widely used. The method described involves altered access points and the opportunity to examine the prosthesis and potentially remove areas of synovial impingement or fibrosis. It represents an opportunity for less-invasive diagnosis and treatment of symptomatic prostheses, but is also in its infancy as a technique, with a higher risk for damaging the prosthesis or neighboring structures due to different access and the instruments required.

### 4.2. Treatment of TMJ Subluxation and Dislocation

TMJ subluxation, or forward displacement of the mandibular condyle past the articular eminence which reduces spontaneously or can be self-reduced, can have multiple different etiologies [98]. It is most commonly an acute event that may be spontaneous or secondary to trauma, a congenital condition, prior dental or otorhinolaryngological procedure, or an underlying psychiatric condition. Rarely, it may become a recurrent or habitual event. It is important to distinguish TMJ subluxation from dislocation, where the condyle is displaced out of the glenoid fossa and usually must be reduced by someone else [98]. Over the years, less invasive procedures have been added to the arsenal of oral and maxillofacial surgeons for recurrent subluxation and dislocation [99].

Botulinum toxin type A, which traditionally has been used to treat focal dystonia or other conditions involving involuntary muscle activity, has recently been shown to be an option for the treatment of recurrent TMJ dislocation. A study by Fu et al. explored the long-term efficacy of botulinum toxin type A for recurrent TMJ dislocation [100]. They found that injections of 25–50 units of BTX-A into the lateral pterygoid muscle were successful in preventing any additional TMJ dislocations during a follow-up spanning 3 months to 2 years, without needing additional injections. However, the sample size of this study was small (*n* = 5), and CTs were obtained to determine the position of the lateral pterygoid muscles. This can be performed with EMG guidance [101] or alternatively could be combined with another procedure with direct or arthroscopic visualization.

Other conservative treatments for habitual TMJ subluxation that have been a focus of research are autologous blood injection (ABI) and dextrose prolotherapy [102]. Studies on ABI indicate that it is effective, although it will occasionally require multiple injections [103,104,105], and has good long-term success [106]. It appears to be more effective when injected in the pericapsular tissues and not just the superior joint space alone [107,108], and the technique may be combined successfully with arthrocentesis or arthroscopy [109,110]. The European Society of TMJ Surgeons (ESTMJS) has recently published a consensus on the treatment of condylar dislocation, finding the best level of evidence for the use of autologous blood as a minimally invasive technique [98]. Following autologous blood prolotherapy, there may be a benefit in limiting the MIO of the jaws. However, intermaxillary fixation alone is currently not recommended [98]. It does appear that a combination of a course of IMF and ABI is more effective at reducing recurrence than ABI alone [105].

Dextrose prolotherapy may have similar effectiveness in the long-term management of symptoms associated with TMJ subluxation. A study by Refai, in which 10% dextrose prolotherapy was administered to 61 patients with symptomatic TMJ subluxation, found a significant reduction and pain, clicking, and frequency of locking in those with symptomatic TMJ subluxation [111]. It is important to note that only three patients in this study had recurrent TMJ dislocation, but they all reported improvement after one treatment. A few systematic reviews evaluating dextrose prolotherapy versus placebo have been performed in recent years. These reviews have found that dextrose prolotherapy significantly reduced pain. However, there were differing findings regarding a significant reduction in MMO and functional scores [112,113]. Additionally, a systematic review by Nagori et al. found that there was no significant difference in the frequency of TMJ subluxation or dislocation [112]. It is important to note that the literature regarding the efficacy of dextrose prolotherapy is not very robust currently, with each systematic review comprising only 3 and 10 randomized controlled trials [112,113]. Thus, it appears that further studies using dextrose prolotherapy would be beneficial in determining its efficacy. A more recent study by Pandey et al. compared autologous blood and 25% dextrose prolotherapy for the treatment of recurrent TMJ dislocation [114]. This retrospective study found that autologous blood prolotherapy was more effective in reducing MMO and improving lateral and protrusive mandibular movements, while dextrose prolotherapy was more effective in reducing pain intensity.

### 4.3. Extended Total Temporomandibular Joint Reconstruction Prostheses (eTMJR)

The first total temporomandibular joint reconstruction procedure was first described in the 1970s. By the early 2000s, multiple companies had developed a total temporomandibular joint prosthesis that consisted of a titanium mandibular condyle and a polyethylene mandibular fossa implant [115]. Although there have been advances in the workflow of conventional total temporomandibular joint reconstruction (TJR) prostheses over the last 20 years, the overall design of the prosthesis replacing the glenoid fossa and the mandibular condyle has largely remained the same. Generally, the mandibular component of the TJR prosthesis does not extend beyond the area of the angle of the mandible. However, in patients with extensive pathologies or deformities involving the TMJ, the conventional prosthesis design cannot be used. For these purposes, extended TJR (eTMJR) prostheses have been designed and used in situations in which defects in the mandible or base of the skull must be reconstructed in addition to the TMJ complex [116,117].

A classification system for the eTMJR prosthesis has recently been described to aid in communication and clinical decision making. There is a separate classification for both the fossa component and the condyle/mandible component. The fossa component classification ranges from F0 (the standard fossa component) up to F5 (fossa prosthesis covering a temporal defect that extends to the jugular foramen). Similarly, the condyle/mandible classification ranges from M0 (standard condyle–ramus component), up to M4 (total alloplastic mandible prosthesis that includes both condyles). This initial classification was based on a review of 19 patients/prostheses from the manufacturer TMJ Concepts (Ventura, CA, USA) [118]. This classification system was recently validated by the same authors by sending a survey to 64 high-volume alloplastic TJR surgeons. It was found that the mandibular component of the classification system had good inter-rater agreement. This was not the case with the fossa classification [119]. This study proposed a revision of the original classification for the fossa component to a simpler three-tier classification system. This classification system ranged from F0 (standard fossa component) to FA (extended fossa component confined to zygomatic arch) to FT (extended fossa component that includes a temporal bone defect). Unfortunately, there were multiple limitations in this study, including the respondent size (*n* = 17) and survey protocol, which made it difficult to view and score the fossa components. The modified fossa component classification was again validated by the same group of authors [120]. This study found better inter-rater agreement with the three-tier fossa component classification system. Again, the study was limited by the small number of respondents (*n* = 24).

Overall, data regarding the efficacy of the various eTMJR prostheses have been limited to primarily case reports and case series [121]. A recent review by Khattak et al. looked to evaluate the effectiveness of eTMJRs based on functional and esthetic variables, and the post-operative complications associated with these prostheses [116]. The variables reported and analyzed included maximum incisal opening, occlusion, symmetry, pain, and diet. The authors found that overall there was an improvement across these variables with the use of eTMJR prostheses. Yet, this study also revealed the significant gaps in information regarding eTMJR prostheses such as the prevalence of post-operative complications (nerve palsy, infection, etc.). Lastly, it was noted that only one of the studies analyzed utilized the eTMJR prosthesis classification system, making it difficult to perform comparative analyses. This highlights the importance of utilizing a classification system and more comprehensive data reporting in future studies on eTMJR prostheses.

## 5. Conclusions

Our knowledge both clinically and surgically regarding the temporomandibular joint has vastly increased over the last 15 years. Landmark studies in basic science, epidemiology, and surgical technique have significantly advanced not only how we treatment plan patients with orofacial pain and TMJ disorders but also the techniques employed to perform procedures that provide the most benefit to the patient. The field continues to evolve as technology improves with novel techniques and prostheses, such as the extended TMJ prosthesis. These advances are allowing TMJ surgeons to not only directly treat the temporomandibular joint, but also the surrounding structures. From patients with JIA to patients with large craniofacial defects, these advancements in knowledge, techniques, and technology are pushing into exciting new territories of TMJ surgery that will continue to revolutionize patient care and quality of life.

## Figures and Tables

**Figure 1 medicina-59-01409-f001:**
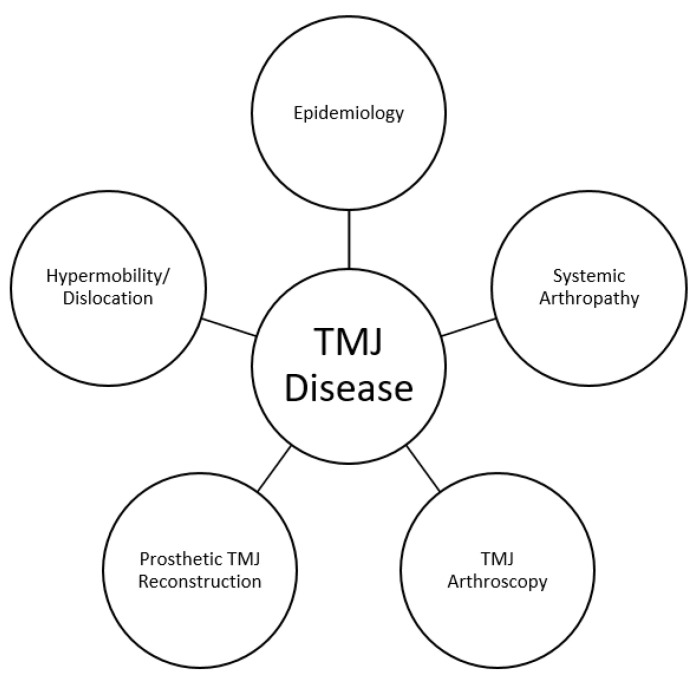
Areas of recent advances impacting surgical management of TMD.

**Figure 2 medicina-59-01409-f002:**
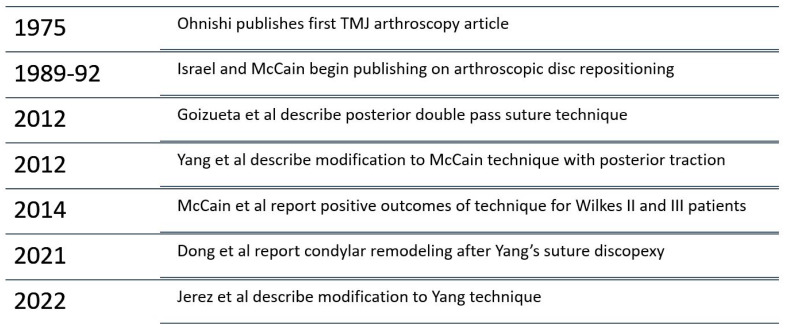
Some highlights of evolution of TMJ arthroscopic discopexy [74,79,80,81,82,83,84,85].

## Data Availability

No new data were created or analyzed in this study. Data sharing is not applicable to this article.

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
