# Peer review of "Recent Advances in Temporomandibular Joint Surgery"

_medicina, 2023, doi:10.3390/medicina59081409_

Round 1

Reviewer 1 Report

This study is a well designed mini review about TMJ surgery. 

Quality of English is sufficient.

Author Response

Dear Reviewer,

Thank you for your review and approval.

Kind regards,

Jonathan Fillmore

Reviewer 2 Report

The manuscript: “Recent Advances in Temporomandibular Joint Surgery” introduces a current and pertinent theme in literature: an update on surgical treatments for temporomandibular disorders. The article is well written, but I have some concerns about the organization and focus of the article:

-          As a reader, I expected that when I went to read about treatments, that would be the article’s primary focus. However, topic 4 is not the most extended and in-depth.

-          This article does not have a story; it seems like a loose information collection. For example, why do the authors include Juvenile Idiopathic Arthritis as the article’s main topic? Wouldn't writing an article on this topic and its treatment would be more interesting?

-          Within the topic of arthroscopy, why does the following topic appear? "4.2 Treatment of TMJ Subluxation and Dislocation" Should it not have been an aborted topic earlier in the manuscript?

-          In topic 4.2, the authors talk about botulinum toxin, botulinum toxin has been used in the treatment of myalgia by several authors (PMID: 37368677; 37104216; 36522255). Shouldn't this also be a topic to be addressed by authors ... how to address chronic myalgia?

-          Why did the authors not address other widely used treatments such as arthrocentesis, open surgery such as discopexy, and discectomy?

-          Overall, the article needs to be reformulated so that the authors include different types of diagnosis in the initial phase of the article, from myalgia, to disc displacement, subluxation, disc perforation, osteoarthritis, etc., then address the different types of treatment, namely botulinum toxin, minimally invasive: arthrocentesis and arthroscopy; open surgery; and finally prosthesis. Authors should indicate which diagnoses and specific therapies are most commonly used.

The article should be revised and corrected in terms of spaces between the references and the text, and some minor grammatical errors.

Reviewer 3 Report

このレビューは、顎関節症とその外科的管理における最近の重要な進歩についての貴重な洞察を提供します。したがって、この論文は出版に値する。

This review provides valuable insight into recent important advances in temporomandibular disorders and their surgical management. Therefore, this paper deserves publication.

Author Response

Dear reviewer,

Thank you for your gracious feedback.

Kind regards,

Jonathan Fillmore MD, DMD

Reviewer 4 Report

This is a very useful and readable review which offers a nice balance between background issues and contemporary TMJ surgery.

I liked the text on headache associated with TMD, which gives a succinct summary of latest thinking on this topic.

I did however feel that the section on the three main psychological groupings could benefit from a brief mention of the role of stressful life events and abnormal illness behaviour - the relevance being case selection for surgery.

Images of TMJ MRI and Cone Beam CTs would help the reader.

I thought that the discussion on arthroscopic versus open discopexy did not comment on the lack of long term data , and the limited amount of medium term follow up data . Does the available evidence favour one approach rather than the other ? Does one open discopexy contribute to the maximum 2 open operations "allowed" by ASTMJS before assigning a TMJ as multiply operated ? Indeed the concept of the multiply operated TMJ was not discussed in the context of potential harms.

Line 296 - missing bracket after   8 patients.

Line 323 - disc , not dixc

ref 40 - inflammation misspelt

Ref 43 - ref incomplete

Refs 101 and 107 - please recheck journal title.

Overall congratulations to the authors for a nice contribution to the literature.

Round 2

Reviewer 2 Report

Dear authors,

Although the authors did not change the manuscript according to all my questions, I was satisfied with the answers and understood the article's purpose.